# ADAR1-mediated regulation of melanoma invasion

Yael Nemlich[1], Erez Nissim Baruch[1,2], Michal Judith Besser[1,2], Einav Shoshan[3], Menashe Bar-Eli[3], Liat Anafi[4], Iris Barshack[4], Jacob Schachter[1], Rona Ortenberg[1] & Gal Markel[1,2,5]

Melanoma cells use different migratory strategies to exit the primary tumor mass and invade surrounding and subsequently distant tissues. We reported previously that ADAR1 expression is downregulated in metastatic melanoma, thereby facilitating proliferation. Here we show that ADAR1 silencing enhances melanoma cell invasiveness and ITGB3 expression. The enhanced invasion is reversed when ITGB3 is blocked with antibodies. Re-expression of wild-type or catalytically inactive ADAR1 establishes this mechanism as independent of RNA editing. We demonstrate that ADAR1 controls ITGB3 expression both at the post-transcriptional and transcriptional levels, via miR-22 and PAX6 transcription factor, respectively. These are proven here as direct regulators of ITGB3 expression. miR-22 expression is controlled by ADAR1 via FOXD1 transcription factor. Clinical relevance is demonstrated in patient-paired progression tissue microarray using immunohistochemistry. The novel ADAR1-dependent and RNA-editing-independent regulation of invasion, mediated by ITGB3, strongly points to a central involvement of ADAR1 in cancer progression and metastasis.

[1] Ella Lemelbaum Institute for Immuno-Oncology, Ramat-Gan 52621, Israel. [2] Sackler Faculty of Medicine, Department of Clinical Microbiology and Immunology, Tel Aviv 69978, Israel. [3] Department of Cancer Biology, MD Anderson Cancer Center, Houston, TX 77030, USA. [4] Department of Pathology, Sheba Medical Center, Ramat Gan 52621, Israel. [5] Talpiot Medical Leadership Program, Sheba Medical Center, Ramat-Gan 52621, Israel. Correspondence and requests for materials should be addressed to G.M. (email: gal.markel@sheba.health.gov.il)

Malignant melanoma is the most aggressive and treatment-resistant form of skin cancer. Melanoma is arguably among the most widely metastasizing neoplastic disease, with a disposition to metastasize as a very early event. Understanding the acquisition of invasive behavior is therefore crucial. One important step for progression to metastatic disease is the transition from radial growth phase (RGP) to the vertical growth phase (VGP)[1]. Specifically, one of the most important proteins associated with melanoma metastatic potential is ITGB3[1–3]. Together with the αV subunit, it forms the heterodimeric adhesion receptor vitronectin. Upregulation of αVβ3 expression occurs in many tissues and has been associated with malignant potential. It is a major cell–extracellular matrix (ECM) mediator that binds a range of ligands containing the amino-acid sequence RGD, mainly collagen, laminin, and fibronectin. Changes in the cytoskeleton organization and altered contacts with the ECM are required for increasing cell motility and intravasation[4,5].

Due to the strong association of ITGB3 with the ability to convert non-invasive RGP melanoma to an invasive VGP melanoma, the biochemical mechanisms that regulate ITGB3 expression in cancer cells are of substantial interest. Experiments with reporter constructs containing regions upstream to the ITGB3 transcription start site show that the transcription factors SP1[6], FoxC2[7], and CDK11P58[8] are involved in the regulation of ITGB3 expression. Additional studies show that miRNAs[9–16] and other regulatory elements, such as protein kinase C[17], activated RAF-MEK-ERK signaling[18], and CCND1b[19] as putative regulators of ITGB3 expression.

RNA editing is a post-transcriptional mechanism through which RNA sequences are directly altered, thus increasing protein diversity from a limited set of genes[20]. The most common form of RNA editing is adenosine-to-inosine (A-to-I) editing, which is catalyzed by members of the family of adenosine deaminases that act on RNA (ADARs) enzymes. In mammals, three ADAR proteins have been identified: ADAR1 and ADAR2 are detected in many tissues; whereas ADAR3 is brain-specific. Rare events of editing in coding regions may result in amino-acid substitutions[21], while editing in non-coding regions might affect splicing, RNA stabilization, and nuclear retention[22]. Furthermore, editing of non-coding RNAs affects their biogenesis or alters their target gene specificity[23,24]. It has been suggested that ADAR plays a role in various biological processes in an RNA editing-independent manner, such: affecting gene expression[25]; processing of miRNA[26–28]; creating protein–protein complexes[29]; and decreasing protein kinase activities[30,31]. The ability to create protein–protein interaction via its double-stranded RNA-binding domain (dsRBD) facilitates ADAR1 to regulate an entire biosythetic pathways directly and systematically[27,28].

We have recently shown that ADAR1 is downregulated along melanoma progression, particularly during the metastatic transition[27], thereby enhancing proliferation[27] and resistance to tumor-infiltrating lymphocytes[32], in an RNA-editing-independent manner. It was shown in a recent seminal paper that ADAR-mediated A-to-I RNA editing occurs in miRNA-455-5p, leading to inhibition of melanoma growth and metastasis in vivo[33]. Here we provide substantial evidence on the role of ADAR1 in melanoma cell invasion by controlling ITGB3 expression independently of RNA editing, at the transcriptional and post-transcriptional levels. These results provide new insights on the mechanistic role of ADAR1 in the acquisition of melanoma metastatic phenotype, as well as on the regulation of ITGB3 expression.

## Results

**ADAR1 controls melanoma cell invasion.** To evaluate the effect of ADAR1 downregulation on the acquisition of invasive potential, four melanoma cell lines (624mel, 003mel, A375, and WM-266-4) were stably transduced with ADAR1-shRNA (knockdown, KD) or non-targeted-shRNA (control), as previously described[27]. These cells represent metastatic (624mel, 003mel, and WM-266) and primary melanoma (A375), express ADAR1, and exhibit basis invasion potential. Expectedly, the constitutive ADAR1-p110 comprised ~90% of total ADAR1 (Fig. 1a, b). Efficient ADAR1-KD was validated for both ADAR1 forms at the mRNA and protein levels using quantitative reverse-transcription PCR (qRT-PCR) and western blot, respectively (Fig. 1a, b). Exposure of the cells to interferon-alpha (IFN-α) induced the ADAR1-p150 but not the ADAR1-p110 (Fig. 1b), confirming that the weak band observed at 150 kD is indeed ADAR1-p150. Matrigel invasion tested both by XTT quantification and by membrane cell fixation, congruently revealed a remarkably enhanced invasion rate following ADAR1-KD in all melanoma cells lines tested (Fig. 1c), confirming the role of ADAR1 in the regulation of melanoma invasion. A strong negative correlation between endogenous ADAR1 expression and invasion activity was demonstrated in 10 different melanoma cell lines (Fig. 1d).

**ADAR1-dependent regulation of invasion is mediated by ITGB3.** We previously published a list of differentially expressed genes following ADAR1-KD in melanoma, and categorized them according to putative function, including invasion (doi:10.1172/JCI62980DS1)[27]. This list of genes (Supplementary Data 1) was analyzed using the online tool String to map potential protein interactions[34]. Importantly, this analysis identifies ITGB3 at the center of the protein network (Supplementary Fig. 1) with a variety of interactions, indicating a key role within this group of altered genes.

ITGB3 is strongly upregulated in melanoma[2] and correlates to the aggressiveness of the tumor[3–5]. On the other hand, we have previously reported that ADAR1 is downregulated upon metastatic transition in melanoma[27]. Remarkably, analysis of ITGB3 and ADAR1 expression levels in 38 low-passage patient-derived metastatic melanoma cultures shows a highly significant negative correlation (Fig. 2a). In line with this observation, the expression of ITGB3 was substantially increased in all four melanoma cell lines after experimental ADAR1 silencing at the mRNA and protein levels (Fig. 2b, c). We therefore hypothesized that the upregulation in ITGB3 expression may explain the enhanced invasiveness following ADAR1 silencing (Fig. 1).

Accordingly, the ability of ITGB3 to interact with the ECM components was blocked with β3 integrin blocking polyclonal antibodies. The four ADAR1-KD or control melanoma cell systems described above (Fig. 1) were pre-incubated for 1 h with 10 μg/ml of the blocking or control antibodies, and then seeded onto the matrigel-coated upper chamber. Invasion rate was evaluated 24 h post seeding. The blocking antibodies significantly decreased the invasion rate of the ADAR1-KD cells as compared to the control antibodies (Fig. 2d and Supplementary Fig. 2). Noteworthy, the blocking antibodies had no significant effect on the invasion rate of the control cells (Fig. 2d). Reduced invasion rates of the control melanoma cells could be observed when a higher concentration of the polyclonal blocking antibodies was used (Fig. 2e and Supplementary Fig. 2). These results suggest that the enhanced invasion driven by ADAR1-KD is mediated by ITGB3.

**ITGB3 expression is directly regulated by miR-22.** ADAR1 was previously demonstrated to affect the expression of many miRNAs[27] while ITGB3 is known to be regulated by several

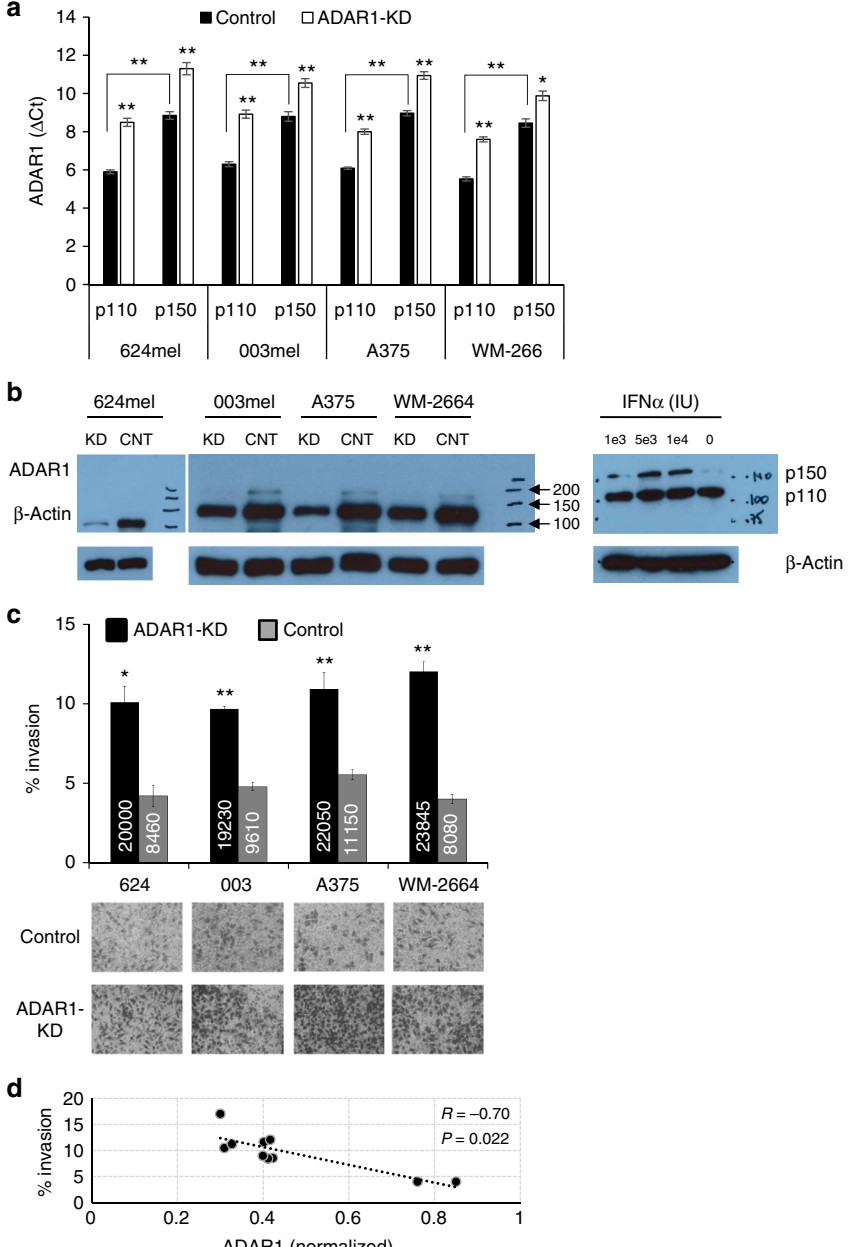

**Fig. 1** Reduced ADAR1 expression leads to increased melanoma invasion. ADAR1 reduced expression in four melanoma cell lines following ADAR1 silencing (ADAR1-KD, KD) or negative control (control, CNT) as determined by **a** qRT-PCR selective for p110 and p150 ADAR1 variants and **b** left: western blot using antibodies against ADAR1 and β-actin as loading control (grouping of images from different gels). Results are representative experiment out of three biologically independent performed; right: western blot for ADAR1 of 624mel cells treated with IFNα. The exponent numbers indicate the amount of international units (IU) used; **c** invasive behavior of ADAR-KD vs. control cell systems (as detailed above). Invasion was performed for 24 h using Boyden chamber assay and monitored by standardized XTT assay or membrane fixation and staining. Error bars indicate ± SEM. Numbers in the bars represent the absolute cell count of invading cells. Representative microphotographs of thincerts are shown; **d** correlation between % invasion and ADAR1 expression quantified by qPCR and normalized to reference melanocytes, in 10 cell lines. Correlation coefficient was determined with Spearman's test. Asterisks represent $P$ values: $*P < 0.05$; $**P < 0.01$ (two-tailed $t$-test)

miRNAs[11,14,15,35,36] and its expression was upregulated following ADAR1-KD (Fig. 2b). Thus, a list of miRNAs predicted to target ITGB3 3′-untranslated region (3′-UTR), based on TargetScan 5.2[37] analysis was crossed with the list of miRNAs, which were downregulated (due to ADAR1-KD) and are known as potential tumor suppressors[27]. This analysis suggests 15 miRNAs: let-7a, miRs-22-3p, -30, 138-5p (miR-138-1 and -138-2), -185-5p, -211-5p, -489-3p, -532-5p, 767-5p, -892b, -938, -1248, -1275, and -1296 as putative candidates as both ADAR1-controlled and ITGB3 regulators (Fig. 3a).

Four miRNAs (miRs-22, -138-5p, -185, and -211) were selected for further examination based on previous studies describing their key involvement in cancer or melanoma invasion[38–46]. None of these miRNAs has been described as a regulator of ITGB3. Accordingly, a portion of ITGB3 3′-UTR containing the putative binding sites for these miRs was cloned. Due to the location of the binding sites and the size of the 3′-UTR, it was divided into two segments UTR-I (putative binding sites for miRs-22-3p, -211-5p, and -138-5p) and UTR-II (putative binding site for miR-185-5p). Both 3′-UTR segments were cloned upstream to Renilla luciferase

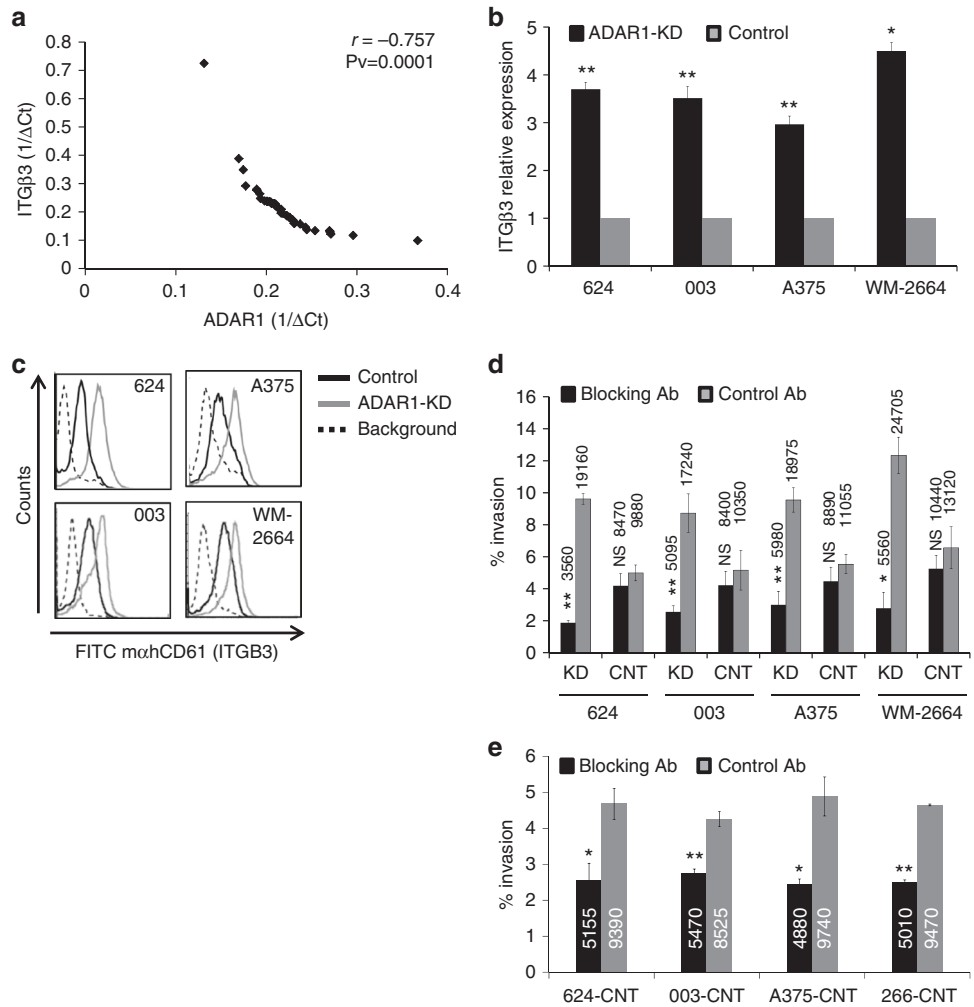

**Fig. 2** ADAR1-regulated invasion is mediated by ITGB3. **a** Normalized ADAR1 and ITGB3 expression level in 38 low-passage primary cultures of metastatic melanoma presented as 1/ΔCt. Correlation was calculated using Pearson test. **b** ITGB3 mRNA levels (qRT-PCR) after silencing of ADAR1 with shRNA (ADAR1-KD) or negative control (control) in four melanoma lines. Results are expressed as fold above negative control. The mean ± SE of three experiments on independent RNA purifications, each performed in triplicates, is shown. **c** Surface levels of ITGB3 expression, tested by flow cytometry in ADAR1-KD or control of each of the indicated cell lines. Background is isotype control; ADAR1-KD increased invasion is inhibited by blocking of ITGB3 function. **d** Stable ADAR-KD (KD) and control (CNT) cell lines or **e** control cell lines only were pre-incubated for 1 h with 10 or 30 μg/ml, respectively, anti-ITGB3 blocking antibody (blocking AB) or control IgG (control AB) and plated on the upper chamber of Boyden chamber assay. The number of invaded cells was evaluated with XTT standardized assay 24 h post seeding. Numbers in the bars represent the absolute cell count of invading cells. Data are presented as the means ± SD from three independent experiments. Statistical significance was determined by Student's *t*-tests. *$P < 0.05$, **$P < 0.01$ (two-tailed t-test)

in a dual luciferase reporting psiCheck2 system. The putative binding sites were altered with three point mutations (UTR-MUT) in one or more binding sites (MUT-A and MUT-B) and a combination of both (MUT-AB), if required, for each miR. All miRs were cloned into the pQCXIP expression vector. Empty psiCheck2 (no UTR) and pQCXIP (Mock) served as negative controls. The various constructs were co-transfected into the easily transfectable HEK 293T cells. The luciferase signal of cells co-transfected with both empty vectors served as point of reference. Forced expression of miRs-22 and -211 with the UTR construct significantly inhibited the luciferase signal while the inhibitory effect was abolished when the UTR-MUT construct was tested (Fig. 3b). On the other hand, no significant difference in luciferase activity was detected for miRs-138-5p and -185-5p. This suggests that miRs-22-3p and -211-5p bind directly to the 3′-UTR of ITGB3. Out of the two potential candidates, which are mostly known as tumor suppressors and as key regulators of invasion[38,40,43,44,47,48], we decided to further investigate miR-22,

mainly due to its unknown role in melanoma and its unique genomic location[49].

To study the effect of miR-22 in melanoma, we used the melanoma cell lines 624, 003, A375, and WM-266-4, which express relatively low levels of endogenous miR-22. The melanoma lines were transiently transfected with mimic-miR-22 oligonucleotides or with control oligonucleotides. MiR-22 was efficiently overexpressed (Supplementary Fig. 3a), while the expression of ITGB3 was significantly reduced both at protein and RNA levels (Fig. 3c–e). This suggests that miR-22 regulates the stability of the mRNA of ITGB3, but inhibition of protein translation cannot be entirely excluded. Functionally, miR-22 significantly inhibited invasion in all tested cell lines (Fig. 3c, d). Importantly, the enhanced ITGB3 expression and invasive function following ADAR1-KD were restored by overexpression of miR-22 in the same cells, as demonstrated in all four melanoma lines (Fig. 3c and Supplementary Fig. 2). Moreover, the reduced ITGB3 expression and invasion function conferred

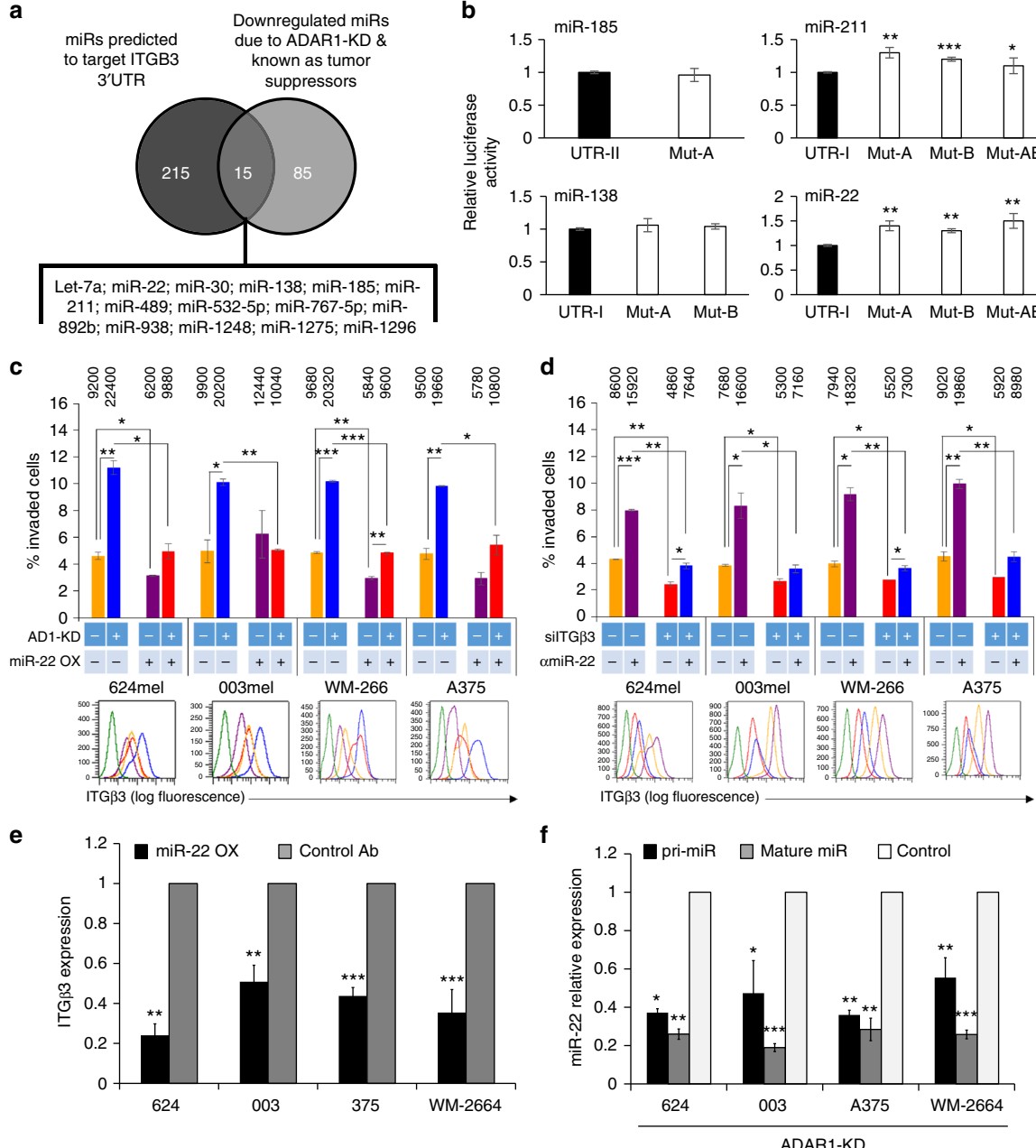

**Fig. 3** ITGB3 expression is directly controlled by miR-22. **a** Venn diagram showing the number of differentially regulated miRNA, which are both known as tumor suppressor, demonstrate reduced expression due to silencing of ADAR1 and predicted to target ITGB3. The 15 identified miRNAs are indicated below the diagram; **b** dual luciferase assays. UTRs (UTR-I or UTR-II) and MUT-UTRs (UTR-MUT-A, UTR-MUT-B, and UTR-MUT-AB) denote ITGB3 3′-UTR segments containing the reference sequence or mutated sequence in the binding site of the respective miR; **c** effect of miR-22 overexpression (miR-22 OX +) or negative control (miR-22 OX−) on the invasion rate and ITGB3 expression in four melanoma lines with (+) or without (−) ADAR1 silencing (AD1-KD). The color code for each treatment scenario is identical for invasion and ITGB3 FACS data; **d** effect of anti-miR-22 (αmiR-22+) or negative control (αmiR-22−) on the invasion rate and ITGB3 expression in four melanoma lines with (+) or without (−) ITGB3 silencing (siITGβ3). The color code for each treatment scenario is identical for invasion and ITGB3 FACS data. Numbers above the bars represent the absolute cell count of invading cells; **e** effect of overexpression of miR-22 or control on mRNA of ITGB3 in four indicated melanoma lines. **f** Expression of both pri- and mature miR-22 form in four ADAR1-KD cell lines, as indicated in the figure, was examined by qRT-PCR. Results represent the mean ± SE of three biologically independent experiments, each performed in triplicates. Asterisks represent P values: *P < 0.05; **P < 0.01; ***P < 0.001 (two-tailed t-test)

by ITGB3-specific KD with siRNA were restored by treatment with anti-miR-22, as demonstrated in all four melanoma lines (Fig. 3d). Similar results were also obtained with stable over-expression of miR-22 cloned into pQCXIP expression vector plasmid (Supplementary Fig. 3b–d). Collectively, these results show that miR-22 directly regulates ITGB3 expression and consequently the invasiveness of melanoma cells.

The expression of pri- and mature forms of miR-22 was tested in all four ADAR1-KD melanoma cell systems with qRT-PCR using form-specific primers. Silencing of ADAR1 led to a reduction in both pri- and mature miR-22 forms (Fig. 3f), suggesting that ADAR1 controls miR-22 expression at the transcription level. Alternatively, post-transcriptional at level of the pri-miR is possible, as it was previously shown that ADAR1

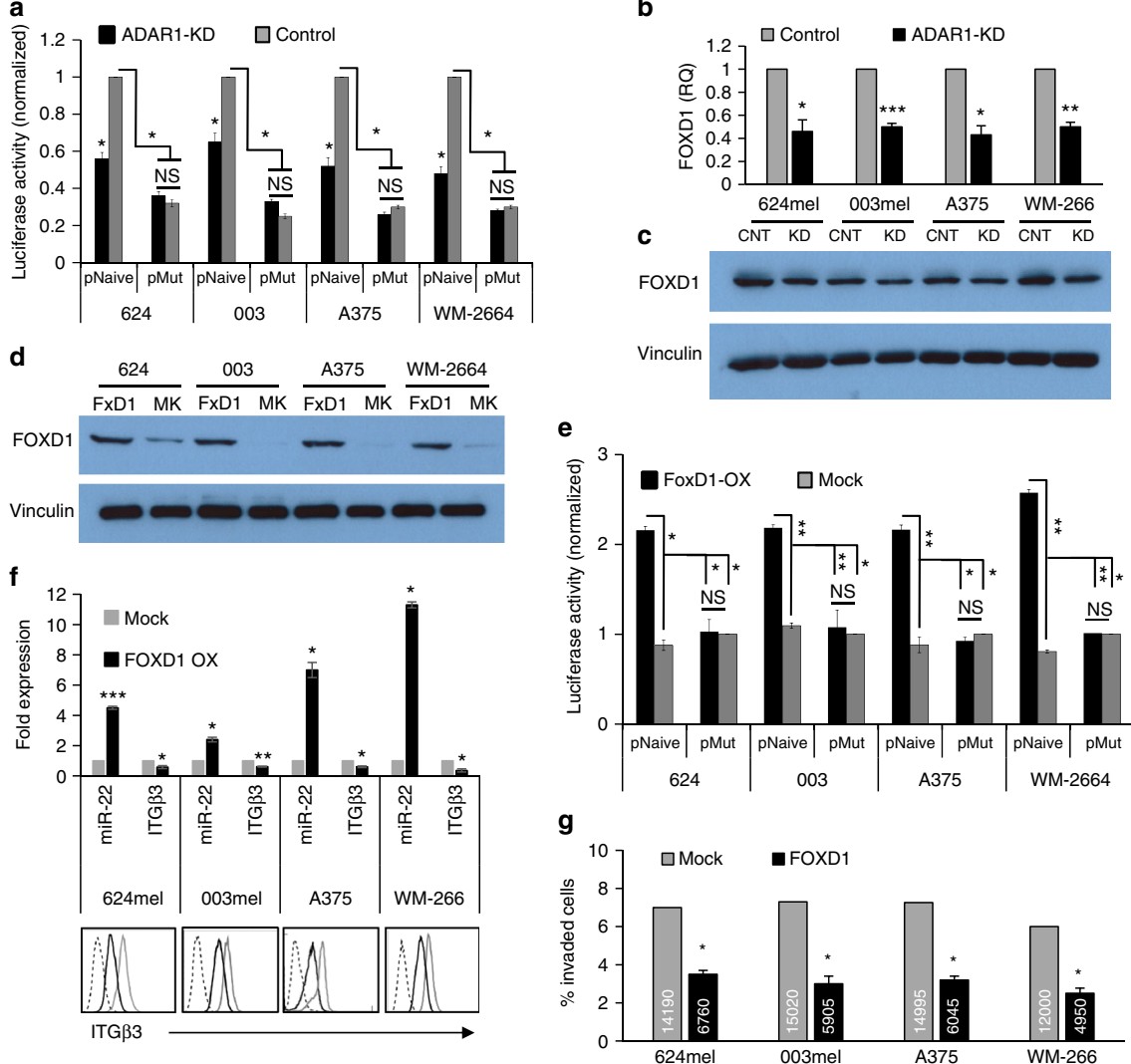

**Fig. 4** FoxD1 overexpression leads to decreased melanoma invasion. **a** The impact of reduced ADAR1 expression (ADAR1-KD, KD) on miR-22 promoter was evaluated by normalized luciferase activity of ADAR1-KD cells as compared to control cells (control, CNT) expressing naive promoter (pNaive) or mutated promoter (pMut) with point mutations at the FoxD1 designate binding site. **b** Expression of FoxD1 in ADAR-KD melanoma cell lines, as indicated in the figure, as compared to control cells was determined by **b** qRT-PCR and **c** western blot. **d** The expression of FoxD1 in four melanoma cell lines, as indicated, that were stably transduced with FoxD1 overexpression plasmid (FoxD1-OX, FxD1) or Mock plasmid (MOCK, MK) was examined by western blot. **e** The impact of increased FoxD1 expression on miR-22 promoter was evaluated by normalized luciferase activity of FoxD1-OX cells as compared to Mock cells expressing naive promoter (pNaive) or mutated promoter (pMut) with point mutations at FoxD1 designate binding site. **f** miR-22 expression was evaluated by qRT-PCR; expression of ITGB3 was determined by qRT-PCR and by extracellular staining with an anti-ITGB3-FITC-conjugated antibody and the corresponding isotype control. **g** Invasive behavior of cells using standardized Boyden chamber assay. The number of cells was evaluated for 24 h post seeding. Numbers in the bars represent the absolute cell count of invading cells. Results for **a**, **b** and **e**–**g** represent the mean ± SE of three biologically independent experiments, each performed in triplicates. Results for **c** and **d** and FACS stains in **f** are representative experiment out of three performed. Asterisks represent *P* values: \**P* < 0.05; \*\**P* < 0.01; \*\*\**P* < 0.001 (two-tailed *t*-test)

can influence miRNA biogenesis by binding to DGCR8, Dicer, or directly to pri-miRs[27,28,33].

**FoxD1 regulates miR-22 expression**. miR-22 is transcribed from exon 2 of a long non-coding gene (host gene)—*miR-22HG*[49]. *MiR-22HG* expression is regulated by transcription factors that specifically bind to their designated sites at the predictive promoter area[50–54]. A segment of 1306 bp (−1262 to +44) of *miR-22HG* promoter[51] was cloned into pGL4.14 luciferase reporter construct and transfected into all four melanoma lines with ADAR1-KD or control. An additional plasmid carrying Renilla gene (pRL) was co-transfected as internal control, and signal

intensity was measured 48 h later. Importantly, the normalized luciferase activity was significantly reduced in all melanoma lines following ADAR1-KD (Fig. 4a). This provides strong positive evidence that the regulation of miR-22 is at the transcription level.

The cloned promoter was analyzed using MAPPER (computational identification of transcription factor-binding sites) to identify putative binding sites for transcription factors. Remarkably, FoxD1 was the only transcription factor common to both the MAPPER-generated list and the list of genes that were downregulated following ADAR1-KD[27]. Importantly, FoxD1 expression was reduced following ADAR1 silencing in all four melanoma ADAR1-KD lines in the mRNA and protein levels

(Fig. 4b, c). It should be noted that the reduction of FoxD1 expression at the protein level is modest, but consistent across cell lines and experiments. Next, we transduced the four parental melanoma cell lines 624, 003, A375, and WM-266-4 with pQCXIP-FoxD1 (FoxD1-OX) or empty pQCXIP (Mock) that served as control. FoxD1 overexpression was confirmed in all cell lines both at the mRNA (Supplementary Fig. 4a) and protein levels (Fig. 4d). The ability of FoxD1 to regulate the promoter of

miR-22HG was tested using the promoter system described above, but with the addition of three neutralizing point mutations that were introduced into the putative FoxD1-binding site (pMut) (Supplementary Fig. 4b). All four melanoma lines (FoxD1-OX or Mock) were transiently co-transfected with the luciferase construct and an additional pRL that served as internal control. Signal intensity measured 48 h later. Supporting the hypothesis that FoxD1 activates the promoter of miR-22HG, luciferase

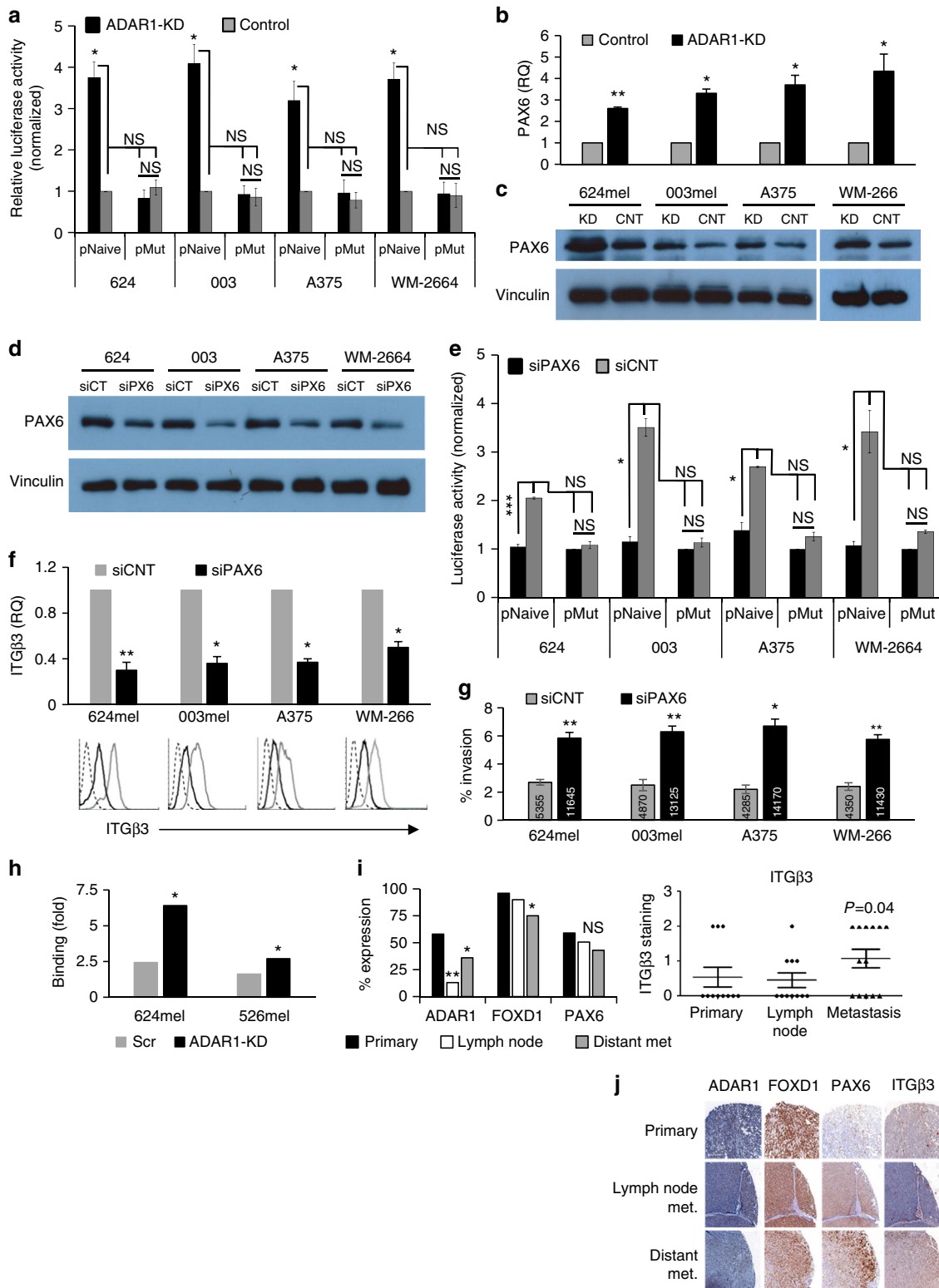

activity was significantly increased when the naive promoter was expressed in the FoxD1-OX cells as compared to Mock cells, without any similar effect when the mutated promoter was tested (Fig. 4e). Indeed, overexpression of FoxD1 enhanced the expression of miR-22 in all four melanoma lines (Fig. 4f). Taken together, these results confirm that FoxD1 enhances miR-22 expression at the transcription level. Finally, we show that overexpression of FoxD1 congruently affects the entire miR-22-dependent chain of events described above, e.g., a reduction in ITGB3 expression in the RNA and protein levels (Fig. 4f) and consequently, the invasiveness of these cells (Fig. 4g and Supplementary Fig. 2).

**ADAR1 regulates ITGB3 expression via PAX6.** As we show the ADAR1 controls the expression level of multiple transcription factors[27], we hypothesized that ADAR1 controls the transcription of *ITGB3*. A segment of 1270 bp (−1120 to +50) of *ITGB3* promoter[6] was cloned into pGL4.14 luciferase reporter construct and transfected into all four melanoma lines with ADAR1-KD or control. An additional pRL was co-transfected as internal control, and signal intensity was measured 48 h later. Importantly, the normalized luciferase activity was significantly increased in all melanoma lines following ADAR1-KD (Fig. 5a). This provides strong positive evidence that ADAR1 controls the transcription of *ITGB3*.

The cloned ITGB3 promoter was analyzed using MAPPER to identify putative binding sites for transcription factors. Remarkably, PAX6 was the only transcription factor common to both the MAPPER-generated list and the list of genes that were upregulated following ADAR1-KD[27]. The increased expression of PAX6 following ADAR1 silencing was verified both in the RNA and protein levels (Fig. 5b, c). Next, the four parental melanoma cell lines 624, 003, A375, WM-26-4 were transfected with PAX6 siRNA (siPAX6) or control siRNA (siCNT). Decreased PAX6 expression was confirmed in all cell lines both in the mRNA (Supplementary Fig. 5a) and protein levels (Fig. 5d). The ability of PAX6 to activate the ITGB3 promoter was tested using the promoter system described above, but with the addition of three neutralizing point mutations that were introduced into the putative PAX6-binding site (p.mut) (Supplementary Fig. 5b). All melanoma lines (siPAX6 or siCNT) were transiently co-transfected with the luciferase construct and an additional pRL that served as internal control. Signal intensity measured 48 h later. Supporting the hypothesis that PAX6 activates the promoter of ITGB3, luciferase activity was significantly decreased when the naïve promoter was expressed in the siPAX6 cells as compared to siCNT cells, without any similar effect when the mutated promoter was tested (Fig. 5e). Indeed, KD of PAX6 congruently

reduced endogenous ITGB3 expression in the mRNA and protein levels (Fig. 5f) and consequently, the invasiveness of these cells (Fig. 5g and Supplementary Fig. 2). Chromatin immunoprecipitation (ChIP) with anti PAX6 or control antibodies was performed on 624mel and 003mel cells (ADAR1-KD or control). ITGB3 promoter sequence includes the PAX6-binding site was quantified by quantitative PCR (qPCR) and normalized to downstream sequence derived from its coding region. Remarkably, significantly higher levels of promoter sequences were measured in the ADAR1-silenced cells as compared to control cells (Fig. 5h). Taken together, these data reveal a new role for PAX6 as a positive regulator of ITGB3 expression and invasion.

The clinical relevance of the ITGB3 regulation pathways by ADAR1 was tested in a patient-paired progression tissue microarray. Importantly, the expression of ADAR1 and FOXD1 decreased, while the expression of ITGB3 increased, along melanoma development. While PAX6 was congruently increased in some patients, it did not reach statistical significance (Fig. 5i, j, Supplementary Fig. 6). This information indicates on the clinical relevance of the ADAR1-ITGB3 pathway in human melanoma progression.

**ADAR1 controls invasion independently of RNA editing.** We have previously described an RNA-editing-independent role for ADAR1 in the control of melanoma cell proliferation by using a system of 624 melanoma cells transfected with: ADAR1-P110 in its wild-type form (ADAR1-OX); ADAR1-P110 bearing specific point mutations in the catalytic site (CAT-MUT-P110) or devoid of the deamination domain (ΔCAT-P110); and empty plasmid as control (Mock)[27]. We focused on ADAR1-P110 due to its dominant constitutive expression (Fig. 1a, b). The reduced A-to-I RNA editing in these mutants was established previously[27]. An ADAR1 construct with neutralizing mutations in all three RNA-binding sites was also created and transfected into 624mel cells. To elucidate whether ADAR1-controlled invasion is dependent or independent of RNA editing, or at least on RNA binding, we tested this system for invasion rate and ITGB3, miR-22 (both pri- and mature), FoxD1, and PAX6 expression levels. Mock cells served as negative control, the OX-P110 cells served as positive controls, and cells transfected with a heterologous RNA-binding protein (Staufen1) served as control for RNA-binding effect. Expression was confirmed by western blot with anti-ADAR1 or anti-Stau1 antibodies (Fig. 6a). ADAR1 levels were quantified with densitometry and normalized to actin levels. Then, ADAR1 levels of all transfectants were normalized to Mock (Fig. 6b). Concurring with the KD experiments, overexpression of ADAR1 (OX-P110) inhibited melanoma cell invasion (Fig. 6c). Importantly, a similar inhibitory effect was observed with the RNA

**Fig. 5** Silencing of PAX6 leads to decreased melanoma invasion. The impact of reduced ADAR1 expression (ADAR1-KD) on **a** ITGB3 promoter was evaluated by normalized luciferase activity of ADAR1-KD (KD) cells as compared to control cells (control, CNT) expressing naive promoter (pNaive) or mutated promoter (pMut) with point mutations at the FoxD1-binding site. Expression of PAX6 in ADAR-KD melanoma cell lines, as indicated in the figure, as compared to control cells was determined by **b** qRT-PCR and **c** western blot; **d** the expression of PAX6 in four melanoma cell lines, transfected with PAX6 siRNA (siPAX6 and siPX6) or control siRNA (siCNT and siCT), was examined by western blot. In **c** and **d** we used antibodies against vinculin as loading control. The impact of decreased PAX6 expression on **e** ITGB3 promoter was evaluated according to normalized luciferase expression in siPAX6 as compared to siCNT-treated cells expressing naive promoter (pNaive) or mutated promoter (pMut) when point mutations were made at the PAX6-binding site; **f** expression of ITGB3 was determined by qRT-PCR and by extracellular staining of ITGB3, followed by flow cytometry analysis. **g** Invasive behavior of the indicated cells using standardized Boyden chamber assay. The number of cells was evaluated for 24 h post seeding. Numbers in the bars represent the absolute cell count of invading cells; **h** chromatin immunoprecipitation for PAX6 followed by qPCR measurements of ITGB3 promoter region that contains the PAX6-binding site and of a control downstream sequence from the ITGB3 coding region. The *Y*-axis denotes the ratio between the qPCR measurements; **i** progression tissue microarray (TMA) was stained the indicated proteins. The staining for nuclear proteins is denoted as percent (bars represent the mean for each progression stage) while surface ITGB3 is denoted as staining intensity (individual cases for each progression stage are shown); **j** staining results of a representative patient. Results for **a**, **b** and **e**–**h** represent the mean ± SE of three biologically independent experiments, each performed in triplicates. Results for **c** and **d** are of a representative experiment out of three performed. TMA included triplicate cores for each case and analyzed with Wilcoxon signed rank test. Asterisks represent *P* values: *$P < 0.05$; **$P < 0.01$; ***$P < 0.001$ (two-tailed *t*-test)

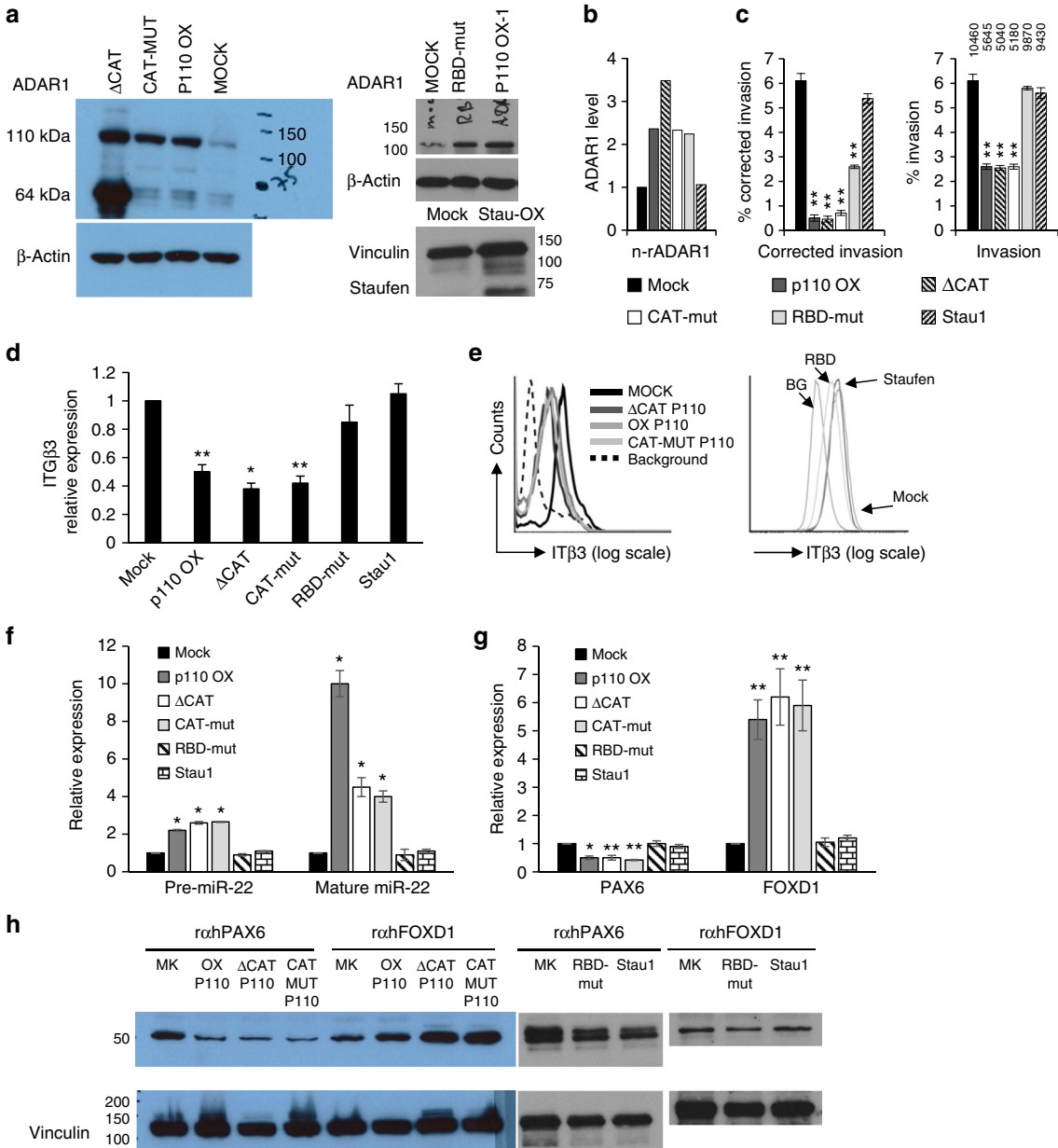

**Fig. 6** ADAR1-dependent regulation of invasion is RNA-editing independent. ADAR1 constructs used for functional assays: OX-P110 (overexpression); ΔCAT-P110; CAT-MUT-P110; and ADAR1 with mutated RNA-binding domains (RBD-mut) and Staufen1 fragments amplified and cloned into pCDNA3 and transfected into 624mel cells. **a** Their expression, relative to MOCK (Control), was detected by western blot (ADAR1 full length is 110 kDa and lacking the catalytic domain the size is reduced to 64 kDa). β-Actin or vinculin served as controls; **b** ratio of normalized ADAR1 in each indicated cell line relative to Mock (n-rADAR1 stands for normalized-ratio). The various transfectants were tested for the following: **c** invasion rate using standardized Boyden chamber assay monitored for 24 h after seeding. Numbers above the bars represent the absolute cell count of invading cells; **d** ITGB3 expression was examined by qRT-PCR and **e** by extracellular staining with an anti-ITGB3-FITC-conjugated antibody over isotype control; **f** pri- and mature miR-22 expression were monitored by qRT-PCR and **g**, **h** FoxD1 and Pax6 expression were evaluated using western blot. Results of **c**, **d** and **f**, **g** represent the mean ± SE of three biologically independent experiments, each performed in triplicates. The numbers in or next to blots indicate molecular marker. Results for **a**, **e**, and **h** are representative of three biologically independent experiments. Asterisks represent $P$ values: $*P < 0.05$; $**P < 0.01$ (two-tailed $t$-test)

editing handicapped CAT-MUT-P110 ΔCAT-P110 cells (Fig. 6c). Similar observations were made with or without correction of migration capacity to ADAR1 expression (Fig. 6c). Inhibition of invasion was corroborated by a reduction in ITGB3, both in the mRNA and protein levels (Fig. 6d, e). This correlated with an increase in miR-22 and FoxD1 expression, as well as with a decrease in PAX6 expression (Fig. 6f–h). Noteworthy, we observed an upregulation in miR-22 expression in OX-P110 cells as compared to CAT-MUT-P110 and ΔCAT-P110. This difference could suggest an effect of A-to-I RNA editing. Indeed, it has

previously been reported that the pri-miR-22 transcript is subjected to A-to-I editing in a number of human and mouse tissues[24]. However, direct sequencing of cDNA segments of pri-miR-22 isolated from Mock, OX-P110, CAT-MUT-P110, and ΔCAT-P110 transfectants, using primers corresponding to the transcript subjected to A-to-I RNA editing, revealed that none of the susceptible adenines were edited (Supplementary Fig. 7). These results are confirmed by another previously published study analyzing high-throughput sequencing of a large set of miRNAs from various human tissues[23]. Importantly, mutations

in the RNA-binding sites completely abrogated the ability of ADAR1 to inhibit invasion (Fig. 6c) or affect the expression of ITGB3, miR-22, FOXD1, and PAX6 (Fig. 6d–h). The corrected invasion capacity of RNA-binding mutant is reduced due to the higher levels of ADAR1 detected by the antibody. Overexpression of another RNA-binding protein, Staufen1, had no effect (Fig. 6b–g). In conclusion, these experiments show that ADAR1 regulates ITGB3 expression and therefore invasion of melanoma cells independently of RNA editing, but the mechanism still depends on the specific RNA-binding capacity of ADAR1.

## Discussion

We have previously reported that downregulation of ADAR1 in melanoma contributes to melanoma growth independently of its RNA-editing activity. Here we show that downregulation of ADAR1 in metastatic melanoma cells causes an increase in ITGB3 expression through RNA-editing-independent transcriptional and post-transcriptional mechanisms, leading to an increase in invasion rate. On the other hand, it was shown that murine models of ADAR1 gene deletion or knock-in of an incapacitating mutation into the catalytic domain, result in lethal autoimmunity[55] associated with aberrant IFN response and facilitated apoptosis mediated by the protein melanoma differentiation antigen 5 (MDA5)[56,57]. MDA5 is an innate dsRNA-sensing molecule that does not have a specific role in melanoma. The discrepancy between these reports and our results may be explained by: (a) The murine models of embryonic lethality may not necessarily point on the same mechanism as established cancer cells. We focused on the biology of malignant cells, which may respond differently than non-malignant cells, especially as malignant cells must inherently overcome mechanisms of apoptosis in order to develop from the first place. (b) The increase in IFN-stimulated genes[56,57] was observed in whole embryos, which include different cell populations with potentially mixed sensitivity to ADAR1 deletion. We analyzed melanoma cell lines, which may be more homogenous with this regar. (c) In the murine studies, ADAR1 function was completely eliminated, while in our studies the ADAR1 was downregulated but still clearly expressed, which might be sufficient to prevent apoptosis. (d) MDA5 is mainly regulated by ADAR1-p150, the inducible, cytoplasmic form of ADAR1. In melanoma, we demonstrate that the dominant form is ADAR1-p110 (Fig. 1), and that the downregulation along melanoma development is indeed mainly of the ADAR1-p110 protein[27]. Noteworthy, it was recently demonstrated in certain cell stress cases, ADAR1-p110 can be exported to the cytoplasm and exert anti-apoptotic effect by inhibiting Staufen1-mediated mRNA decay[58].

In the current study, we identified ITGB3 as mediator of ADAR1 regulation of melanoma cell invasion. We found inverse correlations between ADAR1 and invasion capacity or ITGB3 expression, which is consistent with their previously reported altered expression pattern during melanoma progression from primary to metastatic melanoma, when the cells gain their motile and motility invasive phenotype[2,27]. Accordingly, blocking of ITGB3 ligand-binding site in ADAR1-silenced cells, resulted in reduced invasion. Interestingly, a higher titer of blocking antibodies was required for achieving significant effect on invasion of the control cells, most probably due to stoichiometric differences between the amount of molecules that contribute to the effect and the amount of the blocking antibodies out of the polyclonals. Our data not only outline ADAR1 and ITGB3 interrelations but more importantly demonstrate ITGB3 key role in ADAR1-induced invasion in melanoma.

We provide evidence that ITGB3 is controlled by ADAR1 at transcriptional and post-transcriptional levels, which emphasizes

the potential importance of this process in the acquisition of the invasive phenotype following loss of ADAR1. Post-transcriptional regulation of ITGB3 expression by ADAR1 is mediated by two miRNAs that target directly ITGB3, miR-22, and miR-211 (Fig. 3). As miR-211 is known to have a role in melanoma aggressiveness and migration[40], here we focused on the regulation and role of miR-22. MiR-22 is deregulated in many types of cancers and known to be involved in various cellular processes related to carcinogenesis[38,47,59]. Reduced expression of both pri- and mature miR-22 in ADAR1-silenced melanoma lines was demonstrated, however, as oppose to Luciano at al.[24], and according to our data and previous studies[23,26] no ADAR1-mediated RNA editing of miR-22 precursor was observed. Moreover, previous studies suggest transcriptional regulation of miR-22 by Sp1[53] and AKT[51]. This supports an RNA-editing-independent modulation of miR-22 expression.

ADAR1 interaction with cellular transcription regulatory proteins, both by A-to-I editing of GLI1, a known transcription factor, leading to leading to modulation of transforming growth factor-β signaling pathway[60] and by binding to NF90, a known transcription regulator, via their dsRBDs, causing alterations in gene expression independently of deamination activity. Accordingly, we revealed that ADAR1 silencing alters the expression of two transcription factors—FoxD1 and PAX6, leading to ITGB3 upregulation in two parallel mechanism. FOXD1 is a member of the of the forkhead box (FOX) transcription factors family. Mutated or deregulated FOX genes are often associated with a variety of cancers as tumor suppressors and oncogenes. In addition, it is a strong indicator of successful progression of the gene expression in cell reprogramming[61]. We provide here evidence for a role of FoxD1 in indirect regulation of ITGB3 expression and cell invasion capacity by controlling miR-22 expression (Fig. 4). PAX6 is a member of the Paired Box (PAX) transcription factors family[62] and has been associated with multiple cancer types, either as tumor suppressor or oncogene[63]. Until now, there are no data regarding PAX6 role in melanoma progression, however, it has been strongly linked to a feed-forward regulatory loop with MITF during the onset of melanogenesis[64,65]. We provide here evidence for the direct role of PAX6 in the regulation of ITGB3 transcription, expression, and cell invasion capacity (Fig. 5). The clinical relevance of the ADAR1-ITGB3 pathway was demonstrated in a small cohort of patient-paired melanoma progression tissue microarray. Indeed, statistically significant downregulation of ADAR1 and FoxD1, as well as upregulation of ITGB3, were observed congruently along human melanoma progression (Fig. 5i, j). In line this data, Shoshan et al.[33] has recently demonstrated that ADAR1 inhibits melanoma metastasis in murine models, albeit through a different mechanism, as discussed below. The differences in PAX6 did not reach statistical significance, potentially due to the limited sample size of this small cohort.

Our data indicate that the expression and functional output of both PAX6 and FOXD1 are independent of RNA editing. Indeed, while GLI1 is predicted to be edited in several genomic locations, in addition to the specific non-synonymous editing, leading to its altered function[60], no editing events are predicted[66] or reported for both PAX6 and FOXD1. It is possible that ADAR1 controls their expression via post-transcriptional regulation, as it depends on the RNA-binding activity of ADAR1 (Fig. 6). Therefore, the significant impact of ADAR1 activity either by RNA-editing-dependent or -independent ways on diverse biological process in the cell[67] can be intensified by the regulation of transcription factors that add another layer of complexity to its activity.

We show that ADAR1 regulates ITGB3-mediated invasion and its transcriptional and post-transcriptional regulation independently of RNA editing, but dependently of RNA binding. The

effects exerted by ADAR1 truncation mutants lacking the catalytic domain or with mutated catalytic domain were similar to the full-length ADAR1, while mutated RBD abrogated the effects of ADAR1 (Fig. 6). This points on the importance of the RNA-binding activity of ADAR1 for these functions, which are not exerted by another RNA-binding protein, such as Staufen1 (Fig. 6). Both ADAR1-p110 and Staufen1 can bind 3′-UTR-alu dsRNA[58] and could potentially share other RNA substrates, however they differ strictly by subcellular localization (nuclear and cytoplasmic, respectively), dsRNA-binding dynamics (static and dynamic, respectively), and Z-DNA-binding domains (presence and absent, respectively)[58]. RNA binding is crucial for RNA editing, but regulation of transcription factors such as NF90 by RNA binding independently of RNA editing can occur by creating an RNA bridge, resulting in interactions with other proteins[29]. A recently published study demonstrated a novel role for ADAR1-mediated RNA editing in melanoma progression[33]. This effect of ADAR1 silencing on melanoma tumor growth and metastasis was further confirmed by an in vivo model, suggesting is not confined to invasion in vitro. It was revealed that the editing status of miRNA-455-5p controls melanoma tumor growth and metastasis. Importantly, Shoshan et al. also suggested that ADAR1-mediated regulation of miR-455-5p biogenesis could occur either in an RNA-editing-dependent or -independent manner. While our results demonstrate the RNA-binding-dependent RNA-editing-independent role of ADAR1, the systems we used do not rule out the RNA-editing-dependent mechanisms. Therefore, the combined data suggest a unified model of complex regulation of melanoma cell invasiveness by ADAR1 by multiple mechanisms in multiple layers. For example, here we observed that cells expressing full-length ADAR1 demonstrate higher expression of mature miR-22 as compared to ADAR1-truncated or mutated catalytic domain. While this implies on an editing-dependent effect, no direct editing of miR-22 could be detected, suggesting that this alteration might be due to an indirect effect of RNA editing of additional regulatory elements.

Finally, it was recently shown that ADAR1-p110 can translocate from the nucleus into the cytoplasm under certain cell stress conditions and exert anti-apoptotic effects by inhibiting Staufen1-mediated mRNA decay[58]. The implications of the subcellular localization of ADAR1 on the control of the invasive phenotype and the underlying mechanisms reported here cannot be concluded from this report and require further investigation.

In summary, here we provide substantial evidence for a model, in which ADAR1 controls the expression of ITGB3 in melanoma cells in several distinct RNA-editing-independent mechanisms, and thereby their invasive phenotype. These results complete the previous findings on RNA-editing-dependent roles of ADAR1 in melanoma, setting the stage for a unified contribution to the metastatic phenotype of melanoma cells.

## Methods

**Cells and antibodies**. The melanoma lines 624mel (NCI Surgery Branch, Dr. Steve Rosenberg), A375 (American Type Culture Collection), WM-266-4 (ATCC), G361 (ATCC), WM-115 (ATCC), MeWo (ATCC), MEL-02 (home made[32]), C8161 (ATCC), C81-61 (ATCC), HEK 293T (ATCC), and 003mel (home made[32]) were used. The 38 primary cultures derived from surgically removed metastatic melanoma specimens were established and cultured as described[32]. All cell lines were routinely tested for mycoplasma contamination and were authenticated using mass spec proteomics. Stably transfected cell lines were cultured with 1 μg/ml puromycin (Calbiochem) or 2 mg/ml G418 (Alexis Biochemicals). Incubation of cells with IFN-α (Merck) for 24 h was used to induce the expression of ADAR1-p150.

The following antibodies were used: mouse anti-human-ITGB3 fluorescein isothiocyanate (FITC; BD, Catalog #555753); mouse anti-human-isotype control IgG1 FITC (BD, Catalog #555753); rabbit anti-human ADAR1 (Sigma-Aldrich, SAB4200541); mouse anti-human β-actin (MP Biochemicals Catalog Number: 691001); rabbit anti-human PAX6 (Abcam ab5790); rabbit anti-human Staufen1

(Abcam ab50914); rabbit anti-human FoxD1 (Abcam ab49156); rabbit anti-human ITGB3 (Millipore, AB2984); mouse anti-human ITGB3 (Millipore MAB1957Z); mouse IgG1 (BioXcell, BE0297); and horseradish peroxidase-conjugated secondary antibodies against rabbit IgG or against mouse IgG (Jackson Immunoresearch code 111-035-144).

**RNA isolation and reverse transcription**. Total RNA was isolated using Tri Reagent (Sigma) extraction method. Briefly, the cell pellet first homogenized in Trizol, and then 0.2 ml chloroform/ml Tri reagent was added, samples were centrifuged and the aqueous phase collected. Then 0.5 ml isopropanol/ml Tri reagent was added and the sample was again centrifuged. After discarding the supernatant, the pellet was re-suspended in 75% ethanol, centrifuged, and re-suspended in RNase-free water. Integrity of the RNA was determined by spectrophotometry and electrophoresis. The cDNA pools were generated with a Transcriptor high-fidelity transcriptor kit (Roche) using random hexamer primers or Universal cDNA synthesis kit Exiqon® microRNA cdna kit (Exiqon).

**Real-time quantitative PCR analysis**. Primers (Sigma-Aldrich) were designed according to Primer-Express® software guidelines (Applied Biosystems). Forward and reverse primers were designed from different exons to eliminate possible DNA contamination[27]. miRNA expression was tested with custom Exiqon® primers (Exiqon). The real-time PCR (qPCR) reactions were normalized to GAPDH or U6 endogenous control. Fold of expression was calculated with the accepted ΔΔCt method, as reported previously[27].

**Expression constructs and stable transfections**. The expression systems used in this work were pSuper.puro, pCDNA3.neo, pQCXIP.puro, psiCheck2 (Promega), and pGL4.14 (Promega). The various primers that were designed for cloning and introduction of mutations are described in Supplementary Data 2. Transfections were performed with Turbofect® (Fermentas) according to the manufacturer's instructions. Retroviral transductions were performed as previously described[27]. Site-directed mutagenesis was performed using QuickChange® kit (Stratagene) according to the manufacturer's instructions.

**Anti-miR, oligos, and transient transfection**. 27-mer siRNA oligos specifically targeting PAX6 along with the proper negative control oligos (OriGene) or ITGB3 along with the proper negative control (Dharmacon). Anti-miR-22 oligos along with proper negative control (Dharmacon). The various oligos were transiently transfected (80 nM for siRNAs and 20 nM for anti-miR) with JetPrime® (polyplus) in 96-well microplates, and the cells were tested for miRNA and protein expression 48 h post transfection.

**Western blot**. Lysates of $5 × 10^6$ cells were washed with phosphate-buffered saline (PBS) and lysed in RIPA (Sigma-Aldrich) lysis buffer and protease inhibitor cocktail (Roche) on ice for 20 min. Insoluble material was removed by centrifugation at 14 000 rpm for 10 min at 4 °C. Protein concentration was measured using Pierce™ BCA protein kit (Thermo Scientific). Proteins were separated by 10–12% SDS-polyacrylamide gel electrophoresis, transferred onto nitrocellulose membranes, and incubated with specific antibodies (see Cells and antibodies section). The antigen–antibody complexes were visualized by standard enhanced chemiluminescence reaction (Biological-Industries). Densitometry with ImageJ (NIH) was used for protein quantification.

**Evaluation of RNA editing**. Primers were designed to the genomic sequence in the vicinity of mir-22 sequences in the miRNA registry[68], using NCBI primer design tool[69]. PCR primer design was optimized to give PCR products of approximately 300 bp with at least 100 nucleotides either side of the predicted stem-loop structure. First, PCR was performed on cDNA from three samples of each cell line, then the PCR product was sequenced using additional set of primers. A miRNA was considered to be successfully sequenced if a good-quality sequence of the PCR product was obtained. Sequences were visualized and compared using Chromas (sequence viewer) and NCBI blast, respectively.

**Invasion assay**. Melanoma cells ($2 × 10^5$) were seeded into the upper wells of Transwell invasion system[44] onto Matrigel (BD Biosciences)-coated ThinCerts® PET membranes containing 8-μm pores (Greiner-bio-one) in RPMI 1640 with 0.1% fetal bovine serum (Gibco). In the lower well RPMI 1640 with 10% fetal bovine serum (Gibco) was added. After 24 h of incubation at 37 °C, the cells in the upper well, which didn't invade, were collected, while the number of cells that invaded each membrane was measured by XTT staining as previously described. Percent of invasion was calculated as: (total number of invading cells)/(total number of seeded cells) × 100. The values were adjusted to the relative growth ratio of cells within 24 h evaluated by Net proliferation (standardized XTT), as previously described[46]. In independent experiments, after time allowed for invasion, fluid and cells were removed from upper well and the thincert were fixed and stained with Geimza. The membranes were air dried, removed, and mounted on glass slides. Microphotographs were obtained using bright-field light microscopy (Olympus).

For blocking experiments, 10 or 30 µg/ml of ITGB3 function-blocking antibody or control antibody (see Cells and antibodies section) was added to $2 \times 10^5$ melanoma cells before beginning the assay. Following 1 h of incubation at 4 °C, the cells were seeded in the upper well and the assay was performed as abovementioned. The cells used to examine net proliferation of cells, required for relative growth control, were likewise incubated prior to the beginning of the proliferation assay[70].

**Flow cytometry**. Staining for extracellular antigens was performed on $1 \times 10^5$ cells with the appropriate fluorochrome-conjugated antibodies (see Cells and antibodies section) diluted in fluorescence-activated cell sorting (FACS) medium (PBS, 0.02% sodium azide, and 0.5% bovine serum albumin) on ice for 30 min. Following incubation, cells were centrifuged (5 min, $500 \times g$, 4 °C), washed, and re-suspended in 200 µl FACS medium and collected for FACS analysis. All experiments were performed using a FACSCalibur® instrument (BD Biosciences) and data analysis using FlowJo® software (Tree Star Inc.)[27].

**Luciferase reporter assay**. HEK 293T cells were co-transfected with 1 µg of psiCheck2-ITGB3 3′-UTR (UTR), different psiCheck2-ITGB3 mutated 3′UTR seed sequences (UTR-mutA, UTR-mutB, and UTR-mutAB) corresponding to the miR-binding site(s) or psiCheck2-empty vector (no UTR) and 0.1 µg of the pQCXIP-miRs-22, -211, -138, and -185 or pQCXIP-empty vector (Mock) as control. Cells were harvested 48 h post transfection and assayed with Dual Luciferase Reporter Assay System® (Promega) according to the manufacturer's instructions.

Melanoma cell lines 624mel, 003mel, A375, and wm-266-4 were transfected with 20 µM of siPAX6 or siCNT (control) as previously described (see Anti-miR, oligos, and transient transfection section). Twenty-four hours post siRNA transfection, the cells were additionally transfected with 96 ng of pGL4.14-ITGB3 promoter plasmid (promoter-naive), pGL4.14-ITGB3 mutated promoter (promoter-mut) or pGL4.14-empty vector (control), and 4 ng pRL for evaluating pGL4.14 transfection efficiency. Cells were harvested 48 h post transfection and assayed with Dual Luciferase Reporter Assay System® (Promega) according to the manufacturer's instructions[27].

**Chromatin immunoprecipitation**. ChIP was performed using Pierce Agarose ChIP kit (Thermo Scientific, USA) according to the manufacturer's protocol. Briefly, formaldehyde was added to culture medium at a final concentration of 1% for 10 min at 25 °C, and crosslinking was stopped by incubating in 0.1 M glycine for 5 min. Cells were rinsed with PBS, lysed, and nuclear fraction was isolated. Nuclei were subjected to Micrococcal Nuclease (MNase) digestion (10 U/µl) in 37 °C water bath for 15 min and pelleted by centrifugation; supernatant containing digested chromatin was then collected. Chromatin was incubated with Protein G beads with either 10 µg of rabbit anti-human PAX6 Ab (ab5790; Abcam) or normal rabbit IgG as negative control (supplied with a kit), or rabbit anti-human RNA polymerase II antibody (supplied with a kit) as a positive control, according to the manufacturer's recommendations. After overnight at 4 °C incubation, beads were washed and immunoprecipitation (IP) complex was recovered and treated with proteinase K for 2 h at 65 °C. DNA was recovered using DNA clean up columns supplied with the kit and eluted with 50 µl of PCR-grade water. qPCR amplification was done using 2 µl of ChIP DNA and specific primers for Integrinβ3 promoter PAX6-binding region, Integrinβ3 non-specific downstream gene region, and GAPDH promoter (supplied with a kit). Crosslinked chromatin prior to IP was used as a positive control (input) for PCR amplification.

**Patient-paired tissue microarray**. Progression tissue microarray of paired samples from the same patient was designed in-house. Formalin-fixed, paraffin-embedded paired tissue samples of primary tumors, lymph node metastases, and distant metastases were collected from 12 patients (clinical features in Supplementary Table 1), along with 7 normal liver tissue samples and 3 normal muscle tissue samples, which were used for orientation and control. Each tissue sample was initially stained with hematoxylin and eosin (H&E) and representative areas of tumors were marked by an expert pathologist (I.B.) morphologically. Accordingly, three 2 mm diameter tissue cylinders were punched out from each tumor block and deposited into a recipient block using Manual Tissue Arrayer MTA-1 (Beecher Instruments Inc., Sun Prairie, WI, USA). Tumor sample triplicates were used as a means of overcoming tumor heterogeneity. Post array construction, a 4 µm section was H&E-stained to confirm the histological quality. A consecutive 4 µm section was used for immunohistochemical staining. Each spot was scored by a blinded expert pathologist (I.B.). Uninterpretable cores due to loss of the tissue or excessive background staining were excluded from the analyses. This study was approved by the Institutional Review Board of Sheba Medical Center (Protocol SMC-2406).

**Statistical analysis**. Data were analyzed using the unpaired two-tailed Student's *t*-test. Correlations were examined with Pearson's correlation test. The tissue microarray data were analyzed using Wilcoxon signed rank test. Two-tailed *P*-value ≤ 0.05 was considered significant.

**Data availability**. The authors confirm the availability of the "minimal data set" necessary to interpret, replicate, and build on the findings reported in the paper. Previously generated gene and microRNA datasets in ADAR1-KD cells are available at https://doi.org/10.1172/JCI62980DS1.

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

## Acknowledgements

This work was supported by research grants ISF (Israel Science Foundation) 1489/10 and 1925/15. Special thanks for the Lemelbaum Family, Aronson Foundation, and Allen Berg Fund for Excellence in Immuno-Oncology for their generous support.

## Author contributions

Y.N. and G.M. conceived the study and wrote the manuscript. Y.N. performed the majority of experiments and analyzed data. E.N.B., R.O., L.A., and I.B. performed experiments and analyzed data. M.B. and J.S. provided critical cell lines and reagents. E.S. performed experiments and M.B.-E. critically read and edited the manuscript.

## Additional information

**Competing interests:** The authors declare no competing interests.

