## [Peer Review File · Nature Communications]

Reviewers' comments:

Reviewer #1:

The Nemlich et al manuscript concerns an important and timely topic of cancer biology, tumor invasion and cell migratory mechanisms. The authors provide data from which they conclude that the ADAR1 editing enzyme is involved in melanoma cell invasiveness and ITGB3 expression, but in an editing independent manner, through effects on regulators of ITGB3 expression (miR22, PAX6). Additional evidence is necessary to strengthen and justify the authors' main conclusions.

1. The authors conclude that the ADAR1 mediated effects on invasion are RNA editing independent, for example in figure 6, where they express p110 short form ADAR1, and CAT mutant p110 or deletion CAT p110. However these proteins would all be potent RNA binding proteins as they possess three copies of the RNA binding domain. Two additional controls are necessary: first, an RNA binding mutant of p110 that does not bind RNA; and second, heterologous RNA binding proteins (maybe Staufen and perhaps the ADAR2 cat mut) should be included as controls to show that the observed effects do not simply relate to expression of a strong dsRNA binding protein in the complementation assays.

2. In many of the experiments beginning with figure 1 it is not clear what form of ADAR1 is shown/detected in the western blots or analyzed by qPCR. Pestal et al in Immunity 2015 showed for example that the biology associated with ADAR1 is different for short form p110 and long form p150 ADAR1.

In Figure 1b for example, the authors show only 1 ADAR1 signal in kd and cnt, and no m. wt. markers are included. This problem exists with most westerns in the manuscript. Please show complete gel region that includes both p110 and p150 with appropriate size markers (in fig. 1 and throughout).

Likewise, for the qPCR analyses, please: a. show in methods section the specific primers used for ADAR1; and b. show the results for p110 RNA and p150 RNA separately.

3. In some ways if I understand the authors' data correctly, the results are counter intuitive. It is known from Liddicoat et al and Hartner et al and others that the loss of ADAR1 protein editing activity or protein in the mouse leads to increased apoptosis and an interferon gene expression signature and that occurs because of the loss of editing activity and change in mda5 nucleic acid sensing. Mda5 is Melanoma Differentiation-Associated protein 5. These results should be discussed in the context of the melanoma proliferation findings that the authors report.

4. Many of the results presented in the figures are expressed as a percent (cell invasion rate) or as a relative expression (qPCR). Please show absolute values along with the percent in () for cells in invasion assay, and please give copy number of RNA in the qPCR experiments.

Reviewer #2:

In their study "ADAR1-Mediated Regulation of Melanoma Invasion", Nemlich et. al. demonstrated ADAR1-dependent and RNA-editing independent regulation of invasion mediated by ITGB3. They have shown that ADAR1 downregulates ITGB3 expression at post-transcriptional and transcriptional levels through miR-22 and PAX6 transcription factor, independent of RNA editing. In addition, authors demonstrated that ADAR1 directly regulates miR-22 expression by FOXD1 transcription factor.

This study is interesting and important regarding the role of ADAR1 in acquisition of melanoma metastatic phenotype. However, as major concern, I believe that additional experiments using an

in-vivo assays in mice models and/or clinical relevance of their findings in human samples specimens, will strongly improve the manuscript and is essential for publication.

Minor points:

1. Figure 1: Nemlich et. al. concluded that ADAR1 decreases melanoma cell invasion, as ADAR1 knockdown in 624mel, 003mel, A375 and WM-266-4 melanoma cell lines resulted in increased invasion potential. The authors need to justify in the text why these cell lines were chosen. Particularly WM-266-4 cells which are highly invasive and therefore may not be suitable for this kind of experiments. In addition, to strength their claim, the authors need to examine whether overexpression of ADAR1 in invasive melanoma cells would result in decreased invasion ability. Also, the authors should demonstrate a correlation between ADAR1 endogenous expression and invasion ability in these lines and preferably in more melanoma cell lines. Technically, a representative image of invaded cells should be presented in invasion assays throughout all the text
2. Figure 2: similarly, the authors need to show that overexpression of ADAR1 in high ITGB3 expressing cell lines indeed reduces ITGB3 protein expression level.
3. Figure 3: "293T" cells need to be changed to HEK293T cells. Additionally authors need to explain in the text why all luciferase assays were conducted in these cells, and not in melanoma cells.
4. Figure 3: in order to fully demonstrate the ADAR1-miR-22- ITGB3 axis regulates melanoma invasion, rescue experiments should be perform: 1. miR-22 inhibition effect on cell invasion will be abolish upon cDNA ITGB3 introduction. 2. ADAR1 inhibition effect on cell invasion is diminishing upon anti-miR-22 introduction.
5. Figure 5: authors concluded that ADAR1 regulates ITGB3 expression via PAX6. As this claim entirely relay on luciferase assays, I think that the authors should demonstrate the effect of ADAR1 on the binding efficiency of PAX6 to ITGB3 promoter by EMSA or ChIP assays.

After providing these additional experiments, this paper will be suitable for publication in "Nature Communications"

Point by point response

Reviewer 1

We would like to thank the reviewer for providing perceptive comments, which enhanced the quality of our manuscript.

1. “The authors conclude that the ADAR1 mediated effects on invasion are RNA editing independent, for example in figure 6, where they express p110 short form ADAR1, and CAT mutant p110 or deletion CAT p110. However these proteins would all be potent RNA binding proteins as they possess three copies of the RNA binding domain. Two additional controls are necessary: first, an RNA binding mutant of p110 that does not bind RNA; and second, heterologous RNA binding proteins (maybe Staufen and perhaps the ADAR2 cat mut) should be included as controls to show that the observed effects do not simply relate to expression of a strong dsRNA binding protein in the complementation assays.”

Response:

We would like to thank the reviewer for this insightful comment. As suggested, we added two new controls: ADAR1 with mutations in the three dsRNA binding motifs, and a heterologous RNA binding protein Staufen1. While Staufen1 had no effect on invasion or the expression of any of the proteins (PAX, FOXD1, miR-22 and ITGb3), the effect of ADAR1 was abrogated upon mutation of the RNA binding domain. This suggests that the effect of ADAR1 **depends on its RNA binding** properties and that this effect is not conferred just by any RNA binding protein. ADAR1-p110 and Staufen1 differ by subcellular location, RNA binding dynamics and the presence of Z-DNA binding domain. As the effect is independent of RNA editing, it is hypothesized that other mechanisms that depend on RNA binding, such as RNA bridges, may play a role.

The results are depicted in the new Figure 6. Depiction and explanation for these transfectants are provided in “Results” new page 10 lines 268-270 and 274-275. The results and conclusion are described in “Results” new page 11 lines 293-299. The data is discussed in “Discussion” new pages 14-15 lines 378-392. The primers used for mutating ADAR1 and for cloning of Stau1 are described in Supplementary Table 1.

2. “In many of the experiments beginning with figure 1 it is not clear what form of ADAR1 is shown/detected in the western blots or analyzed by qPCR. Pestal et al in Immunity 2015 showed for example that the biology associated with ADAR1 is different for short form p110 and long form p150 ADAR1. In Figure 1b for example, the authors show only 1 ADAR1 signal in kd and cnt, and no m. wt. markers are included. This problem exists with most westerns in the manuscript. Please show complete gel region that includes both p110 and p150 with appropriate size markers (in fig. 1 and throughout).”

Response:

Indeed, ADAR1-p150 has different cell localization and is inducible by signals such as IFN. In the melanoma cell lines studied here, which were not exposed to IFN, the basal expression level of p150 is very low, prompting us to focus on the constitutive p110.

We agree with the reviewer's comment, and provide the complete gel regions that includes both p110 and p150 with the appropriate size markers in all ADAR1 western blots in the paper (Figure 1b and 6a), as requested. It is now evident that ADAR1-p110 is by far the dominant form.

"Likewise, for the qPCR analyses, please: a. show in methods section the specific primers used for ADAR1; and b. show the results for p110 RNA and p150 RNA separately."

Response:

- a. We added the specific primers used for ADAR1 (common), p110 and p150 to the Supplementary Table 1, which includes the sequences of all of the primers used in this work.
- b. We show the expression data of p110 and p150 in each cell system, before and after ADAR1 knockdown, separately, as the reviewer has requested. In line with the known literature, our previous work (Nemlich et al, J Clin Invest 2013), the protein level data (Fig. 1b) and the tissue level (Fig. 6) - the mRNA of p150 is expressed at around 8-fold lower than the mRNA of p110. ADAR1 knockdown had a similar effect on both ADAR1 variants.

The new results are depicted in the new Fig. 1A, described in Results (new page 4 lines 81-82) and in Discussion new page 12 lines 311-313.

3. "In some ways if I understand the authors' data correctly, the results are counter intuitive. It is known from Liddicoat et al and Hartner et al and others that the loss of ADAR1 protein editing activity or protein in the mouse leads to increased apoptosis and an interferon gene expression signature and that occurs because of the loss of editing activity and change in mda5 nucleic acid sensing. Mda5 is Melanoma Differentiation-Associated protein 5. These results should be discussed in the context of the melanoma proliferation findings that the authors report."

Response:

Following the reviewer's comment, we added several sentences to the Discussion. These appear in the new page 12 lines 305-317. In brief: a) MDA5 is regulated mainly by p150, while in melanoma p110 is dominant; b) We have previously shown that the ADAR1 downregulation in melanoma progression occurs mainly in p110; c) in the murine models mentioned, p150 is deleted/incapacitated entirely, while in melanoma it is downregulated but still expressed; d) p110 can be translocated in certain instances into the cytoplasm to prevent apoptosis.

4. "Many of the results presented in the figures are expressed as a percent (cell invasion rate) or as a relative expression (qPCR). Please show absolute values along with the percent in () for cells in invasion assay, and please give copy number of RNA in the qPCR experiments".

Response:

We agree with the reviewer's request to see the absolute invasion numbers. In line with this request, the other reviewer asked to repeat all invasion experiments and provide microphotographs of the membranes. Therefore, in an attempt to maintain the simplicity of the figures, we added a series of

tables and microphotographs to the supplementary data, to depict the absolute invasion values and raw data of all of the experiments.

The absolute number of invaded cells for all experiments is depicted in Supplementary Table 2.

The qPCR experiments were performed according to standard normalizations, i.e.: each gene in each cell line was normalized according to an internal control, while making sure for experimental validity of reaction efficiency, dissociation and expression range. Data was indeed presented as fold change of the relevant control cells in each experiment. We did not quantify the RNA copy number, as all values were within expression range, and in all cases we further analyzed the changes at the protein level using Western blot (ADAR1, PAX6 and FOXD1) or FACS (ITGB3). As in many of the figures the combined changes in mRNA and protein pointed on transcriptional regulation, this was confirmed by luciferase reporter assays, and in the amended manuscript we also provide CHIP data to confirm this notion. Therefore, we respectfully think that describing the exact change in RNA copy numbers will not provide any new information or further solidify the presented data.

Reviewer #2

We were happy to learn the reviewer considers this study as interesting and important. We would like to thank the reviewer for his insightful comments, which have improved the manuscript significantly.

Major point:

“I believe that additional experiments using an in-vivo assays in mice models and/or clinical relevance of their findings in human samples specimens, will strongly improve the manuscript and is essential for publication”.

Response:

We agree with the reviewer, and think that since all of the models in this manuscript deal with human cell biology, it would be preferable to demonstrate the clinical relevance of our findings in real world human tumor specimens. For that purpose, we developed a melanoma progression tissue microarray (TMA) of paired samples from the same patient, including primary tumor (local disease), lymph node metastasis (regional disease) and distant metastasis. In most cases these archival biopsies were collected at different time points, according to disease progression. Altogether, 13 patients with specimens from the full progression course that were suitable for TMA development were identified in our pathology archives and were included. We would like to emphasize that this is a unique TMA, as other available progression TMAs do not follow the course of disease in individual patients, but rather as a populations. We therefore think that even though the TMA is comprised of a small cohort, its strength is derived from the paired samples, which provide the optimal context to study the clinical relevance of the ADAR1-PAX6-FOXD1-ITGB3 pathway.

Importantly, we show in these paired tissue series that along melanoma progression, the expression of ADAR1 and FOXD1 decrease, while the expression of ITGB3 increases, in a statistically significant manner (Wilcoxon paired test). These results are in line with all of our in vitro data, and therefore provide the clinical relevance in human melanoma development. In certain cases, a clear congruent increase in PAX6 was observed (as shown in Figure 5j). Unfortunately however, we could not observe statistically

significant differences in PAX6 expression in this small cohort. Since the stains for ADAR1, PAX6 and FOXD1 are predominantly nuclear, they are depicted as Percentage. ITGB3 is expressed on surface membrane, and is therefore depicted as Intensity.

The setup of this new tool is described in Methods new page 20 lines 542-558 and the statistical analysis is new page 21 lines 561-562. The data is depicted in the new Figure 5i, described in “Results”, new page 10 lines 255-260 and discussed in “Discussion”, new pages 13-14 lines 360-369. The abstract was amended (page 1 lines 22-23).

Minor points:

Figure 1:

“Nemlich et. al. concluded that ADAR1 decreases melanoma cell invasion, as ADAR1 knockdown in 624mel, 003mel, A375 and WM-266-4 melanoma cell lines resulted in increased invasion potential. The authors need to justify in the text why these cell lines were chosen. Particularly WM-266-4 cells which are highly invasive and therefore may not be suitable for this kind of experiments”.

Response:

We aimed to verify the findings in several melanoma lines available to the scientific community, which would represent metastatic (624mel, 003mel, WM-266) and primary (A375) melanoma. The cells must exhibit measurable invasion potential to enable the effect of various treatments, and be amenable to transfection or transduction.

This is now briefly explained in “Results” new page 4 lines 79-81.

“In addition, to strength their claim, the authors need to examine whether overexpression of ADAR1 in invasive melanoma cells would result in decreased invasion ability”.

Response:

The experiment in Figure 6B demonstrates that overexpression of ADAR1 decreases the invasion ability, as part of investigating the dependence on RNA editing. Following the reviewer’s comment, we now refer to Figure 6B to observe the decreased invasion after ADAR1 overexpression, and hope that it increases the clarity.

“Also, the authors should demonstrate a correlation between ADAR1 endogenous expression and invasion ability in these lines and preferably in more melanoma cell lines”.

Response:

We tested the correlation between ADAR1 endogenous expression and invasion in a total of 10 melanoma cell lines (A375, 624mel, 003mel, WM-266, G361, MeWo, MEL-2, C8161, C81-61 and WM-115). A strong correlation coefficient of -0.70 was observed, which was statistically significant in Spearman's correlation test (two sided p value = 0.022). As point of reference for ADAR1 expression we used normal melanocyte.

The results are depicted in the new Figure 1d and described in "Results" new page 4 lines 89-91. The used cells were added to "Methods", new page 16 lines 417-418, and statistical analysis was added at new page 21 lines 560-561.

"Technically, a representative image of invaded cells should be presented in invasion assays throughout all the text".

Response:

We added representative microphotographs as the reviewer has requested. Most of the figures are highly packed with different panels, therefore in order to maintain the readability of the figures, we added the microphotographs only to Figure 1, while the all of rest are provided in a new Supplementary Figure 2.

We added a new section to "Methods", described in the new page 18 lines 488-492.

These new results are now described in "Results" new page 4 lines 84-85, and depicted in amended Figure 1. Microphotographs in the new Supplementary Figure 2 are referred to at descriptions of all invasion experiments throughout the manuscript.

2. "Figure 2: similarly, the authors need to show that overexpression of ADAR1 in high ITGB3 expressing cell lines indeed reduces ITGB3 protein expression level".

Response:

We agree with the reviewer. Indeed, overexpression of ADAR1 reduces the expression of ITGB3 at the mRNA and protein levels. This data is shown in Figure 6, as part of the analysis of the effects of ADAR1 overexpression, and whether the effect is dependent or independent of RNA editing. Following the reviewer's comment, we now refer to Figure 6b-c to observe the decreased ITGB3 expression after ADAR1 overexpression, and hope that it increases the clarity.

3. "Figure 3: "293T" cells need to be changed to HEK293T cells. Additionally authors need to explain in the text why all luciferase assays were conducted in these cells, and not in melanoma cells".

Response:

We corrected "293T" to HEK 293T, as requested by the reviewer. The luciferase assays were done in order to screen for a microRNA that affects the 3'UTR of ITGb3. In these experiments, portions of the 3'UTR of ITGb3 were isolated by cloning, and relevant point mutations at putative binding sites of microRNAs were introduced. HEK 293T cells were used here because they are easily transfected. These

experiments highlighted miR-22 as the best candidate for further studies. We agree with the reviewer that the effect on melanoma cells and more importantly, on endogenous ITGb3 mRNA and protein expression, should be tested. Indeed, in Figures 3c,d,e we showed the effect of miR-22 overexpression on mRNA, protein and invasion capability of 4 different melanoma lines.

A sentence clarifying the choice of HEK 293T cells was added to “Results”, new page 6 line 145. We further emphasize that the effect of miR-22 was further investigated in melanoma in a sentence added to “Results” new page 6 lines 155-157.

4. Figure 3: in order to fully demonstrate the ADAR1-miR-22- ITGB3 axis regulates melanoma invasion, rescue experiments should be performed: 1. miR-22 inhibition effect on cell invasion will be abolished upon cDNA ITGB3 introduction. 2. ADAR1 inhibition effect on cell invasion is diminished upon anti-miR-22 introduction.

Response

We performed the rescue experiments according to the reviewer’s request. We show that overexpression of miR-22 negates the effect of ADAR1 knockdown, and that the use of anti-miR-22 negates the effect of ITGb3 knockdown. The effect was tested at the mRNA level of ITGb3 and miR-22, ITGb3 protein level and functional level using invasion assays. The experiments were performed in all four cell lines. The new data further confirms all of the causality relations in the ADAR1-miR22-ITGb3-Invasion pathway.

The results are described in Results, new pages 6-7 lines 163-169 and depicted in the new Figure 3c-d and Supplementary Figure 2.

The appropriate additions were made to “Methods” in page 17 lines 456-458.

5. Figure 5: authors concluded that ADAR1 regulates ITGB3 expression via PAX6. As this claim entirely relies on luciferase assays, I think that the authors should demonstrate the effect of ADAR1 on the binding efficiency of PAX6 to ITGB3 promoter by EMSA or ChIP assays.

Response

We conducted ChIP experiments to test the binding of PAX6 to the promoter of ITGB3 but not to a downstream sequence from the coding region. The amount of the ITGB3 promoter (the relevant area of the PAX6 binding site) or downstream sequence were quantified with real time PCR following immunoprecipitation of PAX6. The experiments were performed in ADAR1-silenced 624mel and 003mel cells, as compared to the respective control cells. Importantly, following ADAR1 knockdown, higher levels of ITGb3 promoter sequences were detected following PAX6 immunoprecipitation. Rabbit anti PAX6 antibodies were used as negative control. The input reads were almost identical for each cell type.

The new results are depicted in the new Fig 5h, described in “Results” new page 9 lines 248-253. The “Method” is described in page 19-20 lines 523-541.

After providing these additional experiments, we hope our manuscript will be suitable for publication in “Nature Communications”.

Reviewers' comments:

Reviewer #1 (Remarks to the Author):

The revised manuscript is improved but additional clarifications would further strengthen the manuscript. Regarding the 4 points raised earlier by the first reviewer:

1. Figure 6a expression levels of ADAR1 p110 constructs based on western blots appears very different which complicates the interpretation. ADAR1 protein expression should be quantified, and invasion/function effects expressed relative to ADAR1 protein level.
2. I believe most everyone would agree that p110 short form ADAR1 is the dominant size form expressed and that it is a nuclear protein in a variety of cell types as originally shown about 20 or more years ago. But I do not find a relevant control by the Authors to establish that the larger band that they detect is long form p150 ADAR1. And regarding localization, what is observed by the Authors assays when the NLS of ADAR1 is mutated, so that p110 is cytoplasmic?
3. I find the first paragraph of the Discussion somewhat confusing, in part for the reasons originally noted pertaining to effects of adar1 gene disruption and MDA5 function and enhanced apoptosis.
4. Quantitation of invasion data should be included in the main text, not pushed to supplementary data, as it is central to the main conclusions of the study.

Reviewer #2 (Remarks to the Author):

Authors reply well to all of my comments, as far as I concern paper accepted.

Point by point response

We were very happy to learn that we satisfactorily answered all of the major concerns from the first round. We would like to thank the reviewers for their comments, which have considerably improved the manuscript.

Herein we provide the clarifications requested by Reviewer #1.

1. "Figure 6a expression levels of ADAR1 p110 constructs based on western blots appears very different which complicates the interpretation. ADAR1 protein expression should be quantified, and invasion/function effects expressed relative to ADAR1 protein level".

Response

The expression of ADAR1 was quantified by densitometry in all transfectants and normalized to actin. To standardize the cells, the ADAR1 level in each transfectant was normalized to ADAR1 level in the Mock cells. Altogether, the standardized ADAR1 expression levels seem similar.

This is shown in the new Fig. 6b, and described in Results page 10 lines 274-277 and in Methods page 19 line 487. Figure legends were amended accordingly.

As the reviewer has suggested, invasion was corrected according to the standardized ADAR1 expression by dividing the invasion rate by ADAR1 expression. Lower invasion rates were observed for ADAR1-OX and catalytically inactive ADAR1, regardless of the correction for expression. The raw invasion rate of RBD-mut was similar to that of the Mock cells, indicative that the overexpression of this mutant form of ADAR1 has no effect on invasion. The correction, however, takes into account the higher ADAR1 content (even though it is inactive) and therefore the corrected invasion is lower in RBD-mut than in Mock.

New Fig. 6c shows both invasion and corrected invasion. Results are described in page 10 lines 280-281, and page 11 lines 296-297. Figure legends were amended accordingly.

2. "I believe most everyone would agree that p110 short form ADAR1 is the dominant size form expressed and that it is a nuclear protein in a variety of cell types as originally shown about 20 or more years ago. But I do not find a relevant control by the Authors to establish that the larger band that they detect is long form p150 ADAR1".

Response:

Following the reviewer's comment we treated melanoma cells with IFN α , which is known to induce the expression of p150 but not of the p110. The results of the western blot added to Figure 1b shows that it is the same weak band demonstrated at 150kD under basal conditions that is upregulated after induction with IFN α .

We added a WB panel to Fig. 1b, and sentences to "Results" (page 4 lines 84-85), "Methods" (page 17 lines 438-439) and "Figure Legends".

“And regarding localization, what is observed by the Authors assays when the NLS of ADAR1 is mutated, so that p110 is cytoplasmic?”

Response:

The reviewer points on a highly interesting issue - the effect of ADAR1 localization on the observed phenotype of invasion via ITGb3 expression and function. While we agree with the reviewer that this is a very interesting aspect, we feel that it falls at the boundaries of the research question analyzed and presented in this paper. Indeed, this report doesn't deal with the question of whether it is the nuclear location of p110 that dictates (or not) its effect on miR-22/PAX6/ITGb3/invasion, nor does it make any statement related to this issue. Moreover, we think that conducting experiments only the NLS mutations will not facilitate enough our mechanistic understanding of how p110 controls the ITGb3 pathway. If the effect on ITGb3 will be lost in the NLS mutant, it will not be a surprise that nuclear localization is required for the proper function of a nuclear protein. If the effect on ITGb3 will be retained in the mutant, it will open multiple new mechanistic questions, which divert from the subject of the paper.

We respectfully think that this issue is an entirely new research question that has the potential to become a full manuscript on its own. Therefore, in my opinion these experiments are beyond the scope of the present manuscript,

We now discuss the caveats regarding conclusions that can be made regarding localisation and mechanism in “Discussion” new page 15 lines 419-424.

3. “I find the first paragraph of the Discussion somewhat confusing, in part for the reasons originally noted pertaining to effects of adar1 gene disruption and MDA5 function and enhanced apoptosis”.

Response

Following the reviewer’s comment we amended the discussion. Briefly, the potential explanations for the discrepancy can be explained as depicted herein:

	Murine development	Melanoma
Model studied	Embryo development, fetal liver development	Regulation of established cancer cells
Cell heterogeneity	Multiple heterogenous cell populations	Melanoma cell lines (homogenous)
ADAR1 disruption	Complete/near-complete	Partial
ADAR1 isoform	MDA5 is regulated by ADAR1-p150	P110 is dominant and mediates the effect

We amended this paragraph in the discussion (page 12 lines 303-328).

4. “Quantitation of invasion data should be included in the main text, not pushed to supplementary data, as it is central to the main conclusions of the study”.

Response

Following the reviewer’s comment, we added the absolute cell count of invading cells to all of the appropriate figures and amended the legends accordingly.

Supplementary Table 2 was deleted.

We hope that the revised manuscript satisfactorily addresses all of the comments and is suitable for publications in Nature Communications.

REVIEWERS' COMMENTS:

Reviewer #1 (Remarks to the Author):

The authors responded in a satisfactory manner and addressed the remaining points raised by Reviewer #1 for the first revision. The new figures, the text clarifications, and the organizational changes further improve the manuscript.